



# JuWavelet – Continuous Wavelet Transform and Stockwell-transform for gravity wave analysis

Jörn Ungermann[1] and Robert Reichert[2]

[1]Forschungszentrum Jülich GmbH, Jülich, Germany
[2]Meteorological Institute Munich, Ludwig-Maximilians-Universität, Munich, Germany

**Correspondence:** Jörn Ungermann (j.ungermann@fz-juelich.de)

**Abstract.** This paper describes the Python package JuWavelet, which implements the continuous wavelet transform using the Morlet wavelet, which is a popular tool in the Geosciences to analyse wave-like phenomena. It closes a gap in available software, which are typically focused on discrete transforms or lower dimensions than offered here. The code implements the transform in 1-D, 2-D, and 3-D. In particular, not only the analysis, but also the synthesis from a (modified) decomposition are available. It also provides a consistent implementation for both the original continuous wavelet transform and the derivative Stockwell transform popular in atmospheric gravity wave analysis for all dimensions.

This paper documents the mathematics behind the implementation and offers several examples to showcase the capabilities of the software including the code to generate the shown figures.

## 1 Introduction

Since the introduction of the continuous wavelet transform (CWT) by Grossmann and Morlet (1984), it has been used extensively in the geophysical sciences. In particular the Morlet wavelet, being a harmonic oscillation with a Gaussian envelope is uniquely useful for the analysis of wave-like phenomena. The papers by Stockwell et al. (1996) and Torrence and Compo (1998) further popularized the transform in the atmospheric sciences. Since then, the CWT and the special 'flavour' of CWT introduced by Stockwell labeled Stockwell transform (ST) have been mainstays in the analysis of atmospheric gravity waves (e.g. Kaifler et al., 2015; Chen and Chu, 2017; Ghil et al., 2002; Hindley et al., 2019; Kaifler et al., 2023; Reichert et al., 2024).

Wavelets can be defined in any number of dimensions, but most geophysical applications deal with only 1-D, 2-D, or 3-D data. While literature and sometimes code for 1-D and 2-D wavelets is readily available (e.g. Hindley et al., 2016; Chen and Chu, 2017), in particular code for the 3-D CWT is less accessible.

This paper describes the open source JuWavelet package, which provides 1-D, 2-D, and 3-D implementations in the Python programming language of the CWT using the Morlet wavelet, as well as the associated ST. Earlier versions of this software were used in the publications of (Geldenhuys et al., 2023) and (Krasauskas et al., 2023).

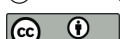



We first introduce the Morlet wavelet and the CWT and its discretisation in 1-D and keep the higher-dimensional
formulas in the appendix for brevity's sake. The intent is to document the mathematical background of the imple-
mentation. A comprehensive and rigorous treatise on wavelets in general can be found in textbooks such as (e.g.
Mallat, 1999), the notation of which we follow here. Lastly, we give some examples on the use of the software pack-
age from our previous work. A particular highlight of the overdetermined CWT is the ability to not only analyse the
data for signals of certain frequencies, but to also extract features from this. This requires the proper implementation
of the inversion of the transform documented below, which this software delivers as well.

## 2 Continuous wavelet transform

This section gives a very brief and mathematical introduction into the CWT and its inversion, here called recon-
struction. The basic idea behind all this is to transform a signal (or more precise function) from its original space
into a higher dimensional space, which spans both the original spatial dimension(s) and, in addition, dimensions
of wavelength/period. This higher-dimensional space then allows for the easier analysis and modification of the
original signal. A comprehensive discussion of this for 1-D is given by Torrence and Compo (1998).

Given an integrable function (typically complex) $\psi$ with zero average and norm of 1, a wavelet dictionary can be
constructed by translations and scaling of $\psi$:

$$\left\{ \psi_{u,s}(t) = \frac{1}{\sqrt{s}} \psi\left(\frac{t-u}{s}\right) \right\}_{u \in \mathbb{R}, s \in \mathbb{R}^+}.$$

These atoms are normalized with $\|\psi_{u,s}\| = 1$. The CWT $Wf$ of the function $f \in L^2(\mathbb{R})$ is then defined as

$$Wf(u,s) = \langle f, \psi_{u,s} \rangle = \int_{-\infty}^{\infty} f(t) \frac{1}{\sqrt{s}} \psi^*\left(\frac{t-u}{s}\right) dt.$$

This convolution can be computed more efficiently on a computer in Fourier-space as

$$Wf(u,s) = \frac{1}{2\pi} \int_{-\infty}^{\infty} \hat{f}(\omega) \sqrt{s} \left[\hat{\psi}(s\omega)\right]^* e^{i\omega u} d\omega,$$

with $\hat{}$ denoting the Fourier transform of a function[1]. The $*$ denotes the complex conjugate.

Let $\psi \in L^2(\mathbb{R})$ be a complex wavelet with

$$C_\psi = \int_0^\infty \frac{|\hat{\psi}(\omega)|^2}{\omega} d\omega < \infty.$$

Then $W$ can be inverted (under certain assumptions on $f$) as

$$f(t) = \frac{2}{C_\psi} \mathrm{Re}\left( \int_0^\infty \int_{-\infty}^\infty Wf(u,s) \frac{1}{\sqrt{s}} \psi\left(\frac{t-u}{s}\right) du \frac{ds}{s^2} \right).$$

---

[1]Here, we follow the convention of $\hat{f}(\omega) = \int_{-\infty}^\infty f(t)e^{-i\omega t}dt$ and $f(t) = \int_{-\infty}^\infty \frac{1}{2\pi}\hat{f}(\omega)e^{i\omega t}d\omega$.





This conserves also the energy of $f$:

$$\int\limits_{-\infty}^{\infty} |f(t)|^2\, dt = \frac{2}{C_\psi} \int\limits_{0}^{\infty} \int\limits_{-\infty}^{\infty} |Wf(u,s)|^2\, du \frac{ds}{s^2}.$$

In practice, a different, simpler reconstruction formula is used. Due to the redundancy of the CWT, also other functions can be used instead of $\psi$ for the reconstruction. Holschneider and Tchamitchian (1991) give the necessary conditions for such a function and the proof. One typically uses the Dirac delta function $\delta$ as this greatly simplifies the computation of the reconstruction formula:

$$f(t) = \frac{1}{C_\delta} \mathrm{Re} \left( \int\limits_{0}^{\infty} Wf(t,s) \frac{1}{\sqrt{s}} \frac{ds}{s} \right)$$

with $C_\delta = \mathrm{Re} \left( \int_0^\infty \frac{\hat{\psi}(\omega)\hat{\delta}(\omega)}{\omega} d\omega \right)$. It has the added advantage that only the wavelet transform evaluated at location $t$ is required to reconstruct $f$ at this location, which is perfect keeping in mind that we can sample the CWT only at discrete locations on a computer.

So, to allow numerical fast execution on computers, we apply discretization. First, we discretize the scales $s$. The data that we analyse is normally sampled and finite and thus band-limited, such that only a finite interval of scales is relevant for the analysis at hand. It is also useful to sample the scales in a logarithmic fashion: $s = s_0 2^{d_j j}$ following Torrence and Compo (1998), with $j \in \mathbb{R}$ for the first step and a natural number in the second step in the following equation. $d_j$ determines the sampling density and is often in the order of $\frac{1}{8}$ to $\frac{1}{2}$. If we assume that only scales between $s_0$ and $s_1 = s_0 2^{d_j J}$, with $J \in \mathbb{N}$ are of relevance, the reconstruction formula becomes thus by substitution

$$f(t) = \frac{1}{C_\delta} \mathrm{Re} \left( \int\limits_{-\infty}^{\infty} Wf(t, s_0 2^{j d_j}) \frac{d_j \ln 2}{\sqrt{s_0 2^{j d_j}}} dj \right) \tag{1}$$

$$\approx \frac{1}{C_\delta} \mathrm{Re} \left( \int\limits_{0}^{J} Wf(t, s_0 2^{j d_j}) \frac{d_j \ln 2}{\sqrt{s_0 2^{j d_j}}} dj \right) \tag{2}$$

$$\approx \frac{1}{C_\delta} \mathrm{Re} \left( \sum_{j=0}^{J} Wf(t, s_0 2^{j d_j}) \frac{d_j \ln 2}{\sqrt{s_0 2^{j d_j}}} \right). \tag{3}$$

This formula can also be used to compute $C_\delta$ more easily as the equation also holds for $f$ being the Dirac delta. With

$$W\delta(u,s) = \frac{1}{2\pi} \int\limits_{-\infty}^{\infty} \hat{\delta}(\omega) \sqrt{s} \left[ \hat{\psi}(s\omega) \right]^* e^{i\omega u} d\omega = \frac{1}{2\pi} \int\limits_{-\infty}^{\infty} \sqrt{s} \left[ \hat{\psi}(s\omega) \right]^* e^{i\omega u} d\omega$$

follows

$$\delta(0) = 1 \approx \frac{1}{C_\delta} \mathrm{Re} \left( \sum_{j=0}^{J} \frac{1}{2\pi} \int\limits_{-\infty}^{\infty} \sqrt{s_0 2^{j d_j}} \left[ \hat{\psi}(s_0 2^{j d_j} \omega) \right]^* e^{i\omega 0} d\omega \frac{d_j \ln 2}{\sqrt{s_0 2^{j d_j}}} \right) \tag{4}$$





$$= \frac{1}{C_\delta} \mathrm{Re} \left( \sum_{j=0}^{J} \frac{d_j \ln 2}{2\pi} \int_{-\infty}^{\infty} \left[ \hat{\psi}(s_0 2^{j d_j} \omega) \right]^* d\omega \right) \tag{5}$$

and thus

$$C_\delta = \sum_{j=0}^{J} \frac{d_j \ln 2}{2\pi} \mathrm{Re} \left( \int_{-\infty}^{\infty} \left[ \hat{\psi}(s_0 2^{j d_j} \omega) \right]^* d\omega \right). \tag{6}$$

## 3 The Morlet wavelet and the Stockwell transform

In the field of geophysics the Morlet wavelet is very popular (e.g. Torrence and Compo, 1998; Chen et al., 2019). Strictly speaking, the Morlet wavelet is not admissible as it has no zero mean, which can be fixed mathematically with some tricks (e.g., using the Heaviside function), but for a discretized analysis of functions of finite support it does not really matter due to the band-limitedness of the involved signals. The code uses the definition in Fourier space:

$$\hat{\psi}_s(\omega) = 2 \frac{\sqrt{s}}{\pi^{0.25}} e^{-\frac{1}{2}(2\pi s\omega - k)^2}.$$

In real space, it is a combination of a Gaussian envelope with a harmonic wave. The free parameter $k$ configures the width of the Gaussian envelope. The Morlet wavelet is shown in Fig. 1 in both spatial and frequency space. It is visible how $k$ widens the Gaussian envelope allowing more periods of the oscillation to fit under it. This directly affects the spectral localization, i.e. more repetitions under the envelope imply a higher spectral resolution and vice versa. The smaller $k$, the larger $\hat{\psi}_s(0)$ becomes, i.e. the "less" admissible it becomes in its unmodified form. Commonly used values are 6 or $2\pi$. The default value for JuWavelet is $2\pi$ as this simplifies the handling of results.

The ST is a variation of the CWT that allows a simpler interpretation of results. The ST $S$ is defined as

$$Sf(u,s) = \sqrt{s} e^{-isu} \langle f, \psi_{u,s} \rangle = \int_{-\infty}^{\infty} f(t) \psi^* \left( \frac{t-u}{s} \right) e^{-isu} dt$$

using the Morlet wavelet with a parameter value of $k = 2\pi$. Using it in this fashion, the wavelet coefficient of scale $s$ can be directly interpreted as amplitude at wavelength $s$ (Stockwell et al., 1996); the proof of this is rather straightforward and will not be repeated here, as it has no relation to the implementation of the software package. This relationship can be proven for an input signal of a harmonic oscillation of infinite extent, but for smaller wavepackets, a dampening may occur. A similar scheme was developed by Chen and Chu (2017) lacking the phase correction. Without phase correction, the complex angle of the coefficients varies by $2\pi$ while been translated over one wavelength. The JuWavelet code implements the pure CWT, the ST, and also the variation introduced by (Chen and Chu, 2017) by name of 'scaled' as this variation only scales the coefficients to align with the amplitude of a infinite monochromatic analysed signal. All relevant transform methods have a *mode* parameter to select the desired





transformation; by default, the Stockwell variation is used, as this delivers the most useful coefficients. Please note that while the normal CWT transforms Gaussian white noise to Gaussian white noise in the coefficients, that the ST introduces a colouring of the noise due to the scaling of coefficients with scale such that coefficients of small periods are more "noisy" than those of large periods. As the difference between the transforms is a linear scaling of coefficients, there is typically no practical effect.

All variations may use any parameter for the Morlet wavelet free parameter $k$. In case a value different from the default $2\pi$ is used, the Wavelet scale parameter of the transform and the period (or wavelength) of the associated wave in the analysed signal differ; the decomposition in the code always returns both the scale and the period for all computed coefficients and the supplied period should be used for almost all purposes. W.r.t. reconstruction, the code contains pre-computed values $C_\delta$ for the most sensible Morlet parameters (i.e. single-digit integers and multiples of

$\pi$); for other values, it will be computed on-the-fly and stored for repeated use.

   The mathematics for the 2-D and 3-D transformation are extensions of the formulas for the 1-D transform and rather verbose due to the added dimensions. They are thus given in the appendices A1 and A2, respectively. They do not introduce relevant new concepts aside the one or two angles rotating the wavelet in two or three dimensions, respectively.

## 115   4   Examples

This section gives several examples demonstrating the power of the available analysis tool.

### 4.1   1-D analysis on synthetic data

The first example serves to showcase the ability of the CWT to analyse a signal for individual components and highlight the differences between the implemented method. Figure 3 shows a signal consisting of three subsignals in

panel (a). The individual signals are shown in panel (e). One signal decreases linearly in amplitude, one changes its frequency and one is only present in the second half, which would be typically time or space in geophysical data. The CWT (panels (b) and (f)) shows nicely in its magnitude the presence and period of the signals. The phase signal is difficult to interpret, as it changes linearly during the signal. Here, we showcase as comparison the result of a Gabor short-time-Fourier-transform (STFT; Allen, 1977; Mallat, 1999, e.g), which here uses a fixed window of length 100.

The transform is similar to the CWT with the Morlet wavelet, the major difference being the Gaussian envelope being replaced by a rectangular boxcar window of length 100. In panels (c) and (g), one can see that the results are similar to the CWT, but that the spectral resolution is better for small periods and worse for large periods. This is a direct consequence of the changing number of periods in the basis function due to the envelope of fixed size. The last panels of (d) and (h) show the result of the ST. The magnitude of the coefficients corresponds to the magnitude

of the original signals, as expected. Another difference can be spotted in the phase plot. The phase stays constant in the analysis at the correct period length, which gives another indication of the "correctly" identified period length.





Due to evaluating only a discrete set of scales on a computer, one seldom really analyses the correct period and is much more likely to look at one "close" to the correct one, where the phase still changes albeit, slowly. Also, many real-life signals vary slightly in frequency, so this makes the phase difficult to interpret even for the ST.

On a more practical note, please observe that the results of the CWT get more "fuzzy" on the edge for larger scales. This is due to the increasing extent of the wavelet basis functions, which extend beyond the original signal, where the algorithm assumes a signal of "zero". This can cause a variety of artifacts in the coefficients close to the border; here, the magnitude decreases, but for data that isn't "zero" at the boundaries, also ringing-like artifacts will appear. To avoid this as much as possible, it is suggested to use tapering, i.e. bringing the signal to zero in a controlled

fashion, and be generally aware of this when interpreting the coefficients. Torrence and Compo (1998) provides a "cone of influence" for 1-D data, which can be computed in the juwavelet.utils module for 1-D decomposition.

### 4.2   1-D analysis of SST

As second example, we'd like to reproduce a figure from Torrence and Compo (1998) analysing an actual series of sea surface temperatures as measure of the amplitude of the El Niño–Southern Oscillation. The software and data

used by Torrence and Compo (1998) for this 1-D analysis is readily available (see Acknowledgements). The analysis is shown in Fig. 2. The time series shows obvious "high-frequency" oscillations of short time scales of a few years plus less obvious long-term features. The power spectrum (i.e. the squared absolute values of the coefficients of the CWT) are shown in Fig. 2b. The power relates to the variance of the original signal explained by the indicated periods and time frames. The hatched region indicates here roughly the coefficients, where the relevant support of

the wavelet extends beyond the region of available data. The third panel c shows the absolute value of the ST, which directly relates to the amplitude in the original signal. While the values are similar (except for the squaring) close to a period of "1", they decrease comparatively for longer periods. The wavelets of JuWavelet are slightly differently normed (real vs. complex integral) such that a scaling factor was necessary for replicating the power spectrum shown in Fig. 2.

### 4.3   2-D analysis on synthetic data

The 2-D Morlet wavelet is shown in Fig. 4 for differing scales and angles. It is used to analyse a synthetic field containing several wave packets depicted in Fig. 5. The synthetic field contains eight partially overlapping waves. All waves have an amplitude of one, but differing envelopes. One example uses a circular pattern similar to gravity waves induced by point sources. The CWT transforms this 2-D field into a 4-D field with added dimensions of

scale (or period) and angle. This 4-D information field is difficult to visualise, so we collapse the 4-D field by selecting for each spatial sample the coefficient with the largest amplitude and select the associated wavelength and angle. This gives again 2-D fields. Figure 5b shows the resulting amplitude field. The amplitudes are all strictly smaller than one, as the relationship between wavelet coefficients and amplitude of original waves holds only for monochromatic waves of infinite extent and the employed examples are all comparatively small. The CWT uses





here the Morlet parameter of $2\pi$. A smaller parameter would deliver more accurate amplitudes at the cost of a
worse spectral resolution. The wavelength associated with the dominant coefficients is depicted in panel (c). The
wavelength of the synthetic wave packets is well reproduced. The direction of the wavefronts is shown in panel (d).
While the amplitudes are almost all underestimated, the identified wavelengths and directions match perfectly with
the employed synthetic waves.

### 4.4   2-D reconstruction

The supplied code can not only compute the decomposition, but also reconstruct the original field from the CWT.
This allows the manipulation of coefficients for a wide variety of purposes. Figure 7 shows several examples thereof.
Please note that the physical dimensions of the data are widely different. While vertically, the measured data covers
about 50 km, horizontally about a thousand are covered. Also the gravity waves in the data have often much larger

horizontal than vertical wavelengths. To facilitate the analysis here, we implemented a simple feature to align the
wavelet basis more with the waves present in the data. The library allows to vertically stretch the data by a con-
stant factor before analysis via an optional *aspect* parameter. Here, an aspect of 40 was employed, which delivered
reasonable results and did not leave the computed coefficients for most angles empty (as is the case for the default
aspect of 1). Using this feature requires some care in computing horizontal and vertical wavelengths (please see

panel titles in App. B).

The data is a down sampled version of the temperature data measured by the ALIMA lidar (Kaifler et al., 2017)
analysed and discussed by Geldenhuys et al. (2023) using an earlier version of this software. The coefficients pro-
vided by the CWT can be filtered, i.e. practically "undesired" coefficients are set to zero before performing a re-
construction. In this manner one can filter out components according to angle and thus separate the original field

into left- (Fig. 7b) or right-slanted waves (Fig. 7c), which here likely corresponds also to downward or upward
propagation direction, respectively. A low pass filter can be finely configured by removing undesired small scales
(Fig. 7d) or vice versa. It is also possible to extract individual wave packets. A simple clustering algorithm contained
in the software package identifies different wave packets and the associated CWT coefficients. By only using these
coefficients in the reconstruction, individual wave packets can be identified and analysed w.r.t. wavelength, direction

and amplitude. The two major features of the field are given in Figs 7e and f.

### 4.5   2-D separation of two mountain waves

In general, gravity waves overlap in the atmosphere. However, most studies focus on analyzing wave parameters
with dominant amplitudes at spatial samples (such as done in Sec. 4.3). In this section, we present a horizontal
cross-section through a vertical wind field created by a numerical simulation. During the simulated flow over a

wide and a narrow mountain, two mountain waves are generated, which overlap. Figure 6a shows the simulated
wave field. We compute the 2-D CWT of the wave field using 58 scales from 4 km to 560 km as well as 9 angles
distributed between 0 and $\pi$ and obtain a 4-D array of wavelet coefficients. Subsequently, we use the watershed



function from the skimage.segmentation package in Python to label the wavelet coefficients. For this, we calculate

the power spectrum and invert it. The watershed function then segments regions in the spectrum that are separated

by a minimum. In this example we find two major segments that are associated with the small- and the large-scale

mountain wave. To reconstruct the small-scale wave, all wavelet coefficients corresponding to the large-scale wave

are set to zero and vice versa. The reconstruction of the small- and large-scale mountain wave is shown in Fig. 6b, and

c. To represent the respective amplitudes, wavelengths, and orientations, we take the adjusted wavelet coefficients

and collapse the modified 4-D array of coefficients by saving the parameters that are associated with the amplitude

maximum at each spatial sample. The result is shown in Figs 6d, e, f, g, h, and i. This example demonstrates the

power of combining the 2-D CWT with an appropriate segmentation algorithm.

### 4.6   3-D analysis on numerical simulation

Figure 8 shows the 3-D spatial structure of a mountain wave simulated with EULAG. It is the same V-shape small-

scale mountain wave that horizontal cross-section is shown in Fig. 6b. The 3-D CWT is applied to the wave field

using 6 azimuth angles distributed between 0 and $\pi$, 7 zenith angles distributed between $-\pi/2$ and $\pi/2$, and 14

scales ranging from $20\,\mathrm{km}$ to $62\,\mathrm{km}$. The aspect ratio is chosen to be 10. As a result of the 3-D CWT we get a 6-D

object containing wavelet coefficients for all three spatial, one scale, and two angle dimensions. In order to illustrate

the amplitude of the wave field, we collapse the 6-D CWT into a 3-D object containing the maximum amplitude

for each spatial sample. We then choose to illustrate the isosurface connected to an amplitude of $0.025\,\mathrm{ms}^{-1}$. The

structure of the amplitude isosurface reveals the discrete nature of the 3-D CWT. Since we use only 6 azimuth and

7 zenith angles, the smoothly varying V-shape structure of the small-scale MW is represented by three 3-D wavelets

that differ only in their azimuth angle.

## 5   Implementation

The software package is available in the Python programming language. It leverages heavily the NumPy library for

its computational needs, which also takes care of leveraging multiple cores if available. As almost all computation

time is spent doing fast Fourier transforms, support for the FFTW3 library (Frigo and Johnson, 2005) is available

via the Python interface pyFFTW. This library offers much better multiprocessing support than the standard numpy

routines. Further parallelization, in particular over multiple nodes of a supercomputer, can be simply realized by

distributing individual scales onto different processes and/or machines and aggregating the results.

## 6   Conclusions

The presented software package closes a gap by providing readily-available 1-D, 2-D, and 3-D CWTs using the

Morlet wavelet. Both analysis and reconstruction formulas are implemented such that also the filtering of signals





for individual wave packets is possible. The software has been used successfully for the published analysis of temperature measurements and is ready to use.

*Code and data availability.* The software is available on Zenodo (Ungermann, 2024) under the AGPL V3 license, where also a link to the git repository can be found.

**Appendix A: Mathematical description of 2-D and 3-D transformations**

**A1 2-D**

The definition for the 2-D Morlet wavelet is

$$\hat{\psi}_{s,\theta}(\omega_x,\omega_y) = \sqrt{8\pi}se^{-\frac{1}{2}\left((2\pi s\omega_x - k\cos(\theta))^2 + (2\pi s\omega_y - k\sin(\theta))^2\right)}$$

with $\theta \in [0,\pi]$ denoting the angle of the wavelet. The transform is given by

$$Wf(u,v,s,\theta) = \frac{1}{(2\pi)^2}\int\limits_{-\infty}^{\infty}\int\limits_{-\infty}^{\infty}\hat{f}(\omega_x,\omega_y)s\left[\hat{\psi}(s\omega_x,s\omega_y)\right]^*e^{i\omega_x u}e^{i\omega_y v}d\omega_x d\omega_y.$$

With the same discretisation of $s$ as in the 1-D case, and a regular sampling of $\theta$ in $\frac{\pi}{K}$ steps, the reconstruction formula is

$$f(x,y) \approx \frac{1}{C_\delta}\sum_{j=0}^{J}\sum_{k=0}^{K}\frac{\pi}{K}\text{Re}\left(Wf\left(x,y,s_0 2^{jd_j},\frac{k\pi}{K}\right)\right)\frac{d_j\ln 2}{s_0 2^{jd_j}}$$

with

$$C_\delta \approx \sum_{j=0}^{J}\sum_{k=0}^{K}\frac{\pi}{K}d_j\ln 2\frac{1}{(2\pi)^2}\text{Re}\left(\int\limits_{-\infty}^{\infty}\int\limits_{-\infty}^{\infty}\left[\hat{\psi}(s_0 2^{jd_j}\omega_x, s_0 2^{jd_j}\omega_y)\right]^*d\omega_x d\omega_y\right).$$

**A2 3-D**

The definition for the 3-D Morlet wavelet is

$$\hat{\psi}_{s,\theta,\phi}(\omega_x,\omega_y,\omega_z) = \frac{4\pi}{\sqrt[4]{\pi}}s^{3/2}e^{-\frac{1}{2}\left((2\pi s\omega_x - k\cos(\phi)\cos(\theta))^2 + (2\pi s\omega_y - k\cos(\phi)\sin(\theta))^2 + (2\pi s\omega_z - k\sin(\phi))^2\right)}$$

with $\theta \in [0,\pi]$ denoting the horizontal turning (azimuth angle) of the wavelet and $\phi \in [-\pi/2,\pi/2]$ the vertical rotation (zenith angle). The transform is given by

$$Wf(u,v,w,s,\theta,\phi) = \frac{1}{(2\pi)^3}\int\limits_{-\infty}^{\infty}\int\limits_{-\infty}^{\infty}\int\limits_{-\infty}^{\infty}\hat{f}(\omega_x,\omega_y,\omega_z)s^{3/2}\left[\hat{\psi}(s\omega_x,s\omega_y,s\omega_z)\right]^*e^{i\omega_x u}e^{i\omega_y v}e^{i\omega_z w}d\omega_x d\omega_y d\omega_z.$$





With the same discretisation of $s$ as in the 1-D case, and a regular sampling of $\theta$ and $\phi$, the reconstruction formula is

$$f(x,y,z) \approx \frac{1}{C_\delta} \sum_{j=0}^{J} \sum_{k=0}^{K} \sum_{l=0}^{L} \frac{\pi}{K} \frac{\pi}{L} \cos(\phi) \mathrm{Re}\left( Wf\left(x,y,z,s_0 2^{jd_j}, \frac{k\pi}{K}, \frac{l\pi}{L}\right) \frac{d_j \ln 2}{(s_0 2^{jd_j})^{3/2}} \right).$$

Please note the factor of $\cos(\phi)$ necessary for integrating the rotation over the unit sphere. The factor $C_\delta$ can be computed as

$$C_\delta \approx \sum_{j=0}^{J} \sum_{k=0}^{K} \sum_{l=0}^{L} \frac{\pi}{K} \frac{\pi}{L} \frac{d_j \ln 2}{(2\pi)^3} \mathrm{Re}\left( \int_{-\infty}^{\infty} \int_{-\infty}^{\infty} \int_{-\infty}^{\infty} \left[ \hat{\psi}(s_0 2^{jd_j}\omega_x, s_0 2^{jd_j}\omega_y, s_0 2^{jd_j}\omega_z) \right]^* d\omega_x d\omega_y d\omega_z \right).$$

**Appendix B: Example code**

This section shows the code used for the 2-D example of Fig. 7 as an example of the ease of use. A more comprehensive version can be found in the software in 'examples/decompose2d.py'. The code for Fig. 3 is given in the file 'examples/decompose1d.py'. The code for Fig. 2 is given in the file 'examples/sst.py'. The code for Fig. 5 is given in the file 'examples/separate2d.py'.

```
     import os
import matplotlib.pyplot as plt
     import numpy as np

     from juwavelet import transform, utils

     storage = np.loadtxt(os.path.join(os.path.dirname(__file__), "alima.txt"))
     xs, ys, wave = storage[0, 1:], storage[1:, 0], storage[1:, 1:].T
     xs -= xs[0]

dx = np.diff(xs).mean()
     dy = np.diff(ys).mean()
     cwt = transform.decompose2d(
         wave, dx=dx, dy=dy,
         s0=20, dj=0.25, js=20, jt=18, aspect=40)
     amps, idxs, iwave = utils.identify_cluster2d(
         cwt, min_amp=2.0, thr=1.0)
```



```
    decomposition, period, theta = [
cwt[_x] for _x in ["decomposition", "period", "theta"]]

    orig = decomposition.copy()

    fig, axs = plt.subplots(2, 3)
axs = axs.T
    opts = {"cmap": "RdBu_r", "vmin": -5, "vmax": 5, "rasterized": True}

    axs[0, 0].set_title("original")
    axs[0, 0].pcolormesh(xs, ys, wave.T, **opts)
    decomposition[:] = orig
    decomposition[:, (np.pi / 2 < theta)] = 0
    rec = transform.reconstruct2d(cwt)
    axs[1, 0].set_title("left slanted")
axs[1, 0].pcolormesh(xs, ys, rec.T, **opts)

    decomposition[:] = orig
    decomposition[:, (theta < np.pi / 2)] = 0
    rec = transform.reconstruct2d(cwt)
axs[2, 0].set_title("right slanted")
    axs[2, 0].pcolormesh(xs, ys, rec.T, **opts)

    decomposition[:] = orig
    decomposition[period < 100, :] = 0
rec = transform.reconstruct2d(cwt)
    axs[0, 1].set_title("low pass")
    axs[0, 1].pcolormesh(xs, ys, rec.T, **opts)

    decomposition[:] = orig
decomposition[iwave != 1] = 0
    idx = idxs[1]
    rec = transform.reconstruct2d(cwt)
    axs[1, 1].set_title(
```





```
        f"$\lambda_x$={period[idx[0]]/np.cos(theta[idx[1]]):3.0f}km "
f"$\lambda_z$={period[idx[0]]/(cwt['aspect']*np.sin(theta[idx[1]])):3.1f}km")
    axs[1, 1].pcolormesh(xs, ys, rec.T, **opts)
    decomposition[:] = orig
    decomposition[iwave != 7] = 0
    idx = idxs[7]
rec = transform.reconstruct2d(cwt)
    axs[2, 1].set_title(
        f"$\lambda_x$={period[idx[0]]/np.cos(theta[idx[1]]):3.0f}km "
        f"$\lambda_z$={period[idx[0]]/(cwt['aspect']*np.sin(theta[idx[1]])):3.1f}km")
    axs[2, 1].pcolormesh(xs, ys, rec.T, **opts)
    for ax in axs[:, 1]:
        ax.set_xlabel("distance (km)")
    for ax in axs[0, :]:
        ax.set_ylabel("altitude (km)")
    plt.show()
```

*Author contributions.* JU wrote the software and the mathematical part of the paper. RR contributed examples and user expertise. JU and RR reviewed and edited the whole paper.

*Competing interests.* The authors declare that they have no conflict of interest.

*Acknowledgements.* We acknowledge Natalie Kaifler (DLR) for providing the ALIMA data used in one of the examples as well as Michael Binder (DLR) for providing the EULAG simulations. Python 1-D wavelet software provided by Evgeniya Predybaylo based on Torrence and Compo (1998) was used for validation; it is available at "http://atoc.colorado.edu/research/wavelets/".





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



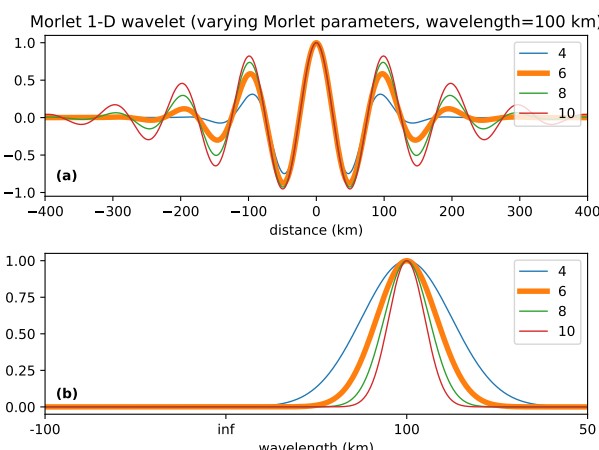

**Figure 1.** The Morlet wavelet in spatial and frequency space. The Morlet parameter influences the number of wave crests within the envelope and thus the localization in frequency space.



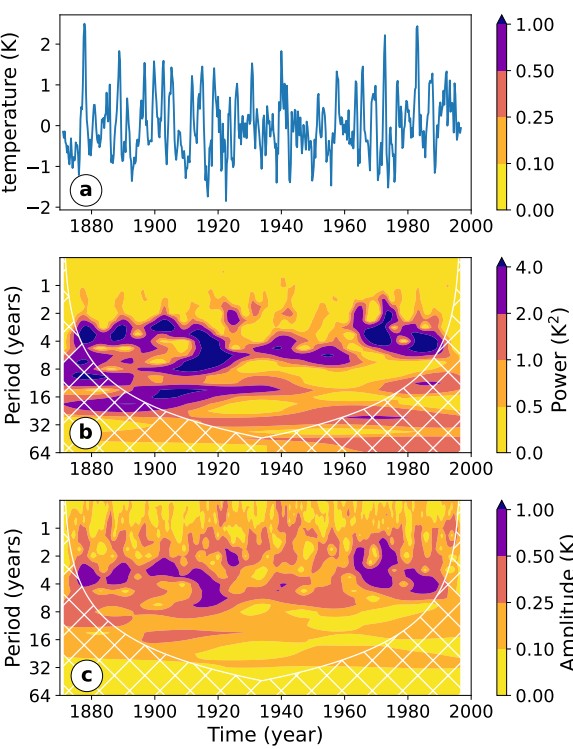

**Figure 2.** A replication of the analysis of the Niño3 SST index from Torrence and Compo (1998). Panel **(a)** shows the index itself, while panel **(b)** shows the power spectrum of the 1-D CWT analysis using a Morlet wavelet (with parameter $2\pi$). The hatched region indicates untrustworthy values where the relevant wavelet support extends over the region where data is available. Panel **(c)** shows the amplitudes derived from the ST for comparison.

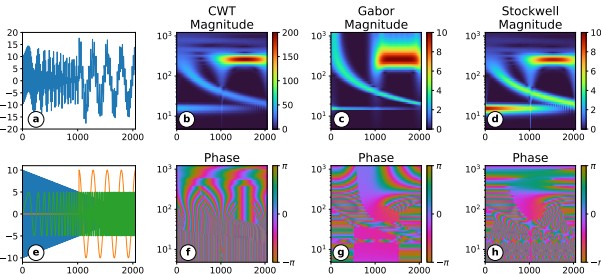

**Figure 3.** An analysis of a combination of three signals, which each vary in a different quantity (amplitude, frequency, location). Panel (a) shows the assembled signal while panel (e) shows the individual signals. Panel (b) and (f) show the magnitude and the phase of the CWT coefficients. Panel (c) and (g) show the magnitude and phase of the Gabor STFT coefficients. Panels (d) and (h) show the magnitude and phase of the ST coefficients.





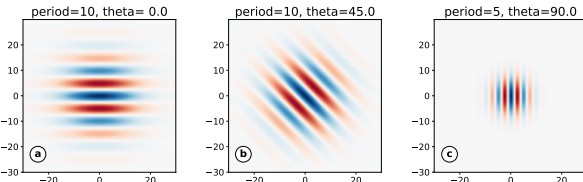

**Figure 4.** The 2-D Morlet wavelet. The three panels showcase variations when changing the rotation or the scale parameter.

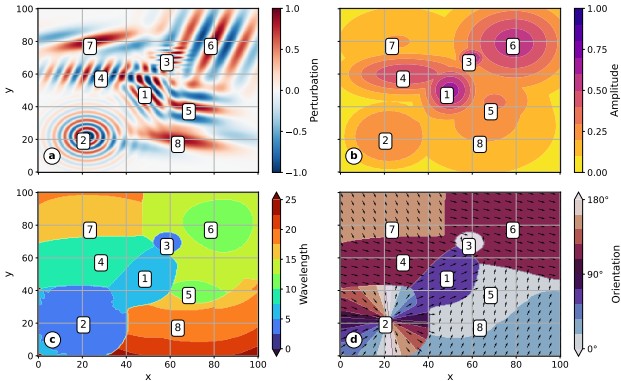

**Figure 5.** An example of a 2-D CWT analysis. Panel (a) shows the synthetic field containing several waves. A CWT analyses this field and the "dominant" wavelet, i.e., the wavelet with the largest coefficient is identified for each spatial sample. Panel (b) shows the resulting amplitude, panel (c) the corresponding wavelength and panel (d) the corresponding orientation.

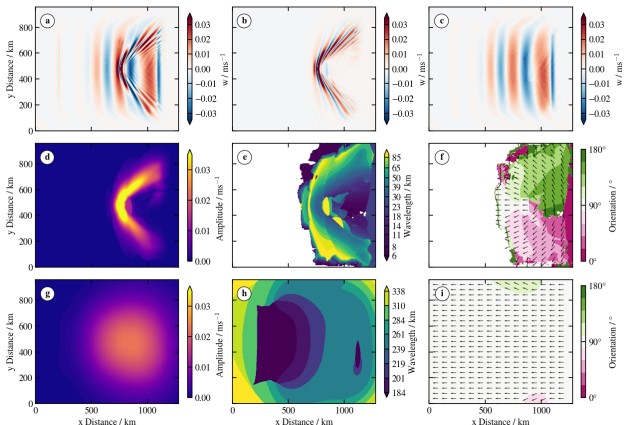

**Figure 6.** This example shows the capabilities of reconstructing a data set from a modified set of coefficients. Panel (a) shows a simulated superposition of two MWs, one of large-scale and one of small-scale. Panels (b) and (c) show filtered reconstructions of the small-scale and large-scale MWs. The sum of (b) and (c) gives the original data. Panels (d), (e), and (f) show the amplitude, wavelength, and orientation of the small-scale MW. Panels (g), (h), (i) show the amplitude, wavelength, and orientation of the large-scale MWs.



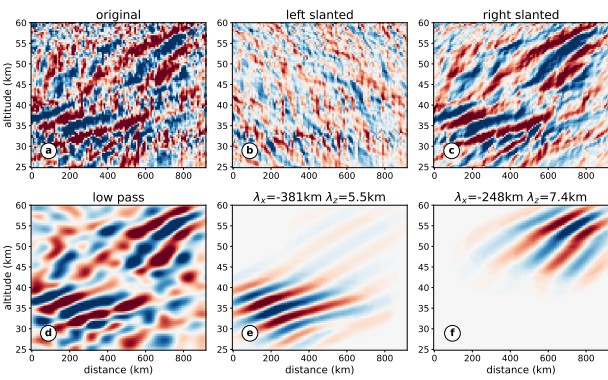

**Figure 7.** This example shows the capabilities of reconstructing a data set from a modified set of coefficients. Panel (a) shows the original data field. Panel (b) shows a filtered reconstruction allowing only left slanted waves while panel (c) shows the right slanted waves. The sum of (b) and (c) gives the original data. Panel (d) shows the effect of discarding the small scales, effectively a low pass. Panels (e) and (f) show reconstructions containing only coefficients from the vicinity of two local maxima in the CWT corresponding to two wave packets, which can be identified in this manner with wavelengths of 381 km horizontally and 5.5km vertically for panel (e) and 248 km horizontally and 7.4 km vertically for panel (f).

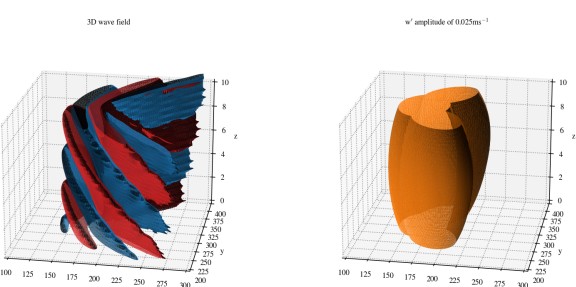

**Figure 8.** An example of a 3-D CWT analysis. The left panel shows a simulated mountain wave field. Isosurfaces are drawn for $w = \pm 0.04\,\text{ms}^{-1}$. A CWT analyses this field and the "dominant" wavelet, i.e., the wavelet with the largest coefficient is identified for each spatial sample. The right panel shows the isosurface for an amplitude of $0.025\,\text{ms}^{-1}$.