# Peer review of "JuWavelet – Continuous Wavelet Transform and S-transform for wave analysis"

_Geoscientific Model Development, 2024_

## Referee Comment (RC1)

**Comments on the paper "JuWavelet – Continuous Wavelet Transform and Stockwell-transform for gravity wave analysis" by Jörn Ungermann and Robert Reichert submitted for publication in GMD ($Paper\ gmd - 2024 - 207$)**

First of all, I would like to congratulate the authors for developing a new and relevant tool for data analysis. Since this software package is freely available, many scientists will use it in the future. Consequently, the paper describing the application of this method needs to be more extensive and detailed.

In particular, the paper needs to provide details on all the parameters employed (for example, $s0$, $dj$, $js$, $jt$, *aspect*, etc.). For instance, in chapter 4.4, the authors introduce an optional *aspect* parameter which is set to 40 in their example. How am I supposed to determine this coefficient if my data, for example, consists of time on the x-axis and altitude on the y-axis?

Essential to this work are not only well-presented examples, but also clear instructions on how to apply this method to any new data.

A plain text explaining the algorithm is also desirable.

Last but not least, an explanation of the results obtained is important.

For example, Figure 7e shows some results of decomposition. The obtained horizontal wavelength is 381 km and vertical wavelength is 5.5 km. However, these parameters are not constant in time and altitude. Consider reconstructed fluctuations at 35 km altitude (Figure 1, blue line). Horizontal wavelength is shorter at the beginning than at the end. For orientation, a dashed line with $\lambda_x =$381 km is shown. In the range of 0 to approximately 500 km $\lambda_x < 381$ km and $\lambda_x > 381$ km after 500 km. At an altitude of 50 km ( compare orange and red dashed lines) $\lambda_x < 381$ km.

Considering vertical cuts at 200 km and 600 km distances, an increasing $\lambda_z$ is noticeable above around 40 km and below around 30 km (see Figure 2).

[Figure]

Figure 1. Perturbations at 35 km and 50 km altitudes are taken from Figure 7e of the manuscript. Dashed lines indicate $\lambda_x$ =381 km.

[Figure]

Figure 2. Perturbations at distances of 200 km and 600 km. Dashed lines indicate $\lambda_z$ =5.5 km.

Waves with defined fixed parameters can be reconstructed using the inverse 2D-FFT. See examples in Fig. 3 c-f. A broader range of $\lambda_x$ and $\lambda_z$ is needed

[Figure]

Figure 3. a,b: Figure7 (a, e) is taken from the manuscript. c-f: 2D-FFT reconstructed fluctuations with parameters as labeled in the titles. )

for the explanation of results from Figure 7 of the manuscript. Hopefully, after a clear description of the method, the interpretation of results will be more obvious.

This comment addressed a specific example. The manuscript, however, requires the authors to offer a broader explanation of potential outcomes, demonstrating its applicability not just through examples but also for users working with different data.

Minor comment: Almost all examples shown in the manuscript can be found in the software package "juwavelet-v01.00.00/examples/", with the exception of Figures 6 and 8. Could these examples also be added to the repository?

---

## Referee Comment (RC2)

**Review of gmd-2024-207: "JuWavelet – Continuous Wavelet Transform and Stockwell-transform for gravity wave analysis" by Jörn Ungermann and Robert Reichert**

**Overview**

In this study, the authors introduce and describe a new Python package called JuWavelet, which implements both the 1-D, 2-D and 3-D continuous wavelet transform (CWT) using the Morlet wavelet and also the widely-used S-transform. These tools are widely used in the geoscience for, among many other applications, the analysis of atmospheric gravity wave perturbations in a variety of geophysical datasets, such as those from satellite, radar and other remote sensing techniques.

By creating this simple and efficient software package, the authors have provided a valuable tool for the community, doubly so because it is coded in the open-source Python language. Their description of the analysis processes and its mathematical foundations are well articulated and their test applications are thorough. The figures are of very high quality representing 1-D, 2-D and 3-D test cases on synthetic, modelled and observed geophysical datasets.

Overall I recommend this study for publication after the authors have considered my comments below, which are minor but I think they would improve the paper by more accurately clarifying technical aspects, introducing some of the geophysical concepts, better referencing existing methods and literature, and helping the interpretation of results by future readers.

I have "General Comments" and "Specific Comments" which it would be nice if the authors could respond to, and then "Minor Typos and Suggestions" for which I do not require a response.

**General Comments**

- Journal Scope: I notice that this article is submitted to Geophysical Model Development, which at first
  was a little surprising to me as it is perhaps better suited to Atmospheric Measurement Techniques?
  However, I will leave the editor to decide whether this is the right EGU journal and I certainly do not
  want to trigger another review cycle with AMT for the authors, as the paper is, in my view almost ready
  for acceptance.
- 2. The paper describes an analysis method specifically focused on the analysis of gravity waves in geophysical datasets, but doesn't actually mention what gravity waves are, how they are typically detected, why we would want to measure them, or any of the atmospheric science aspects of gravity wave remote sensing, detection or analysis. This is fine if the paper is *only* describing a generalised wavelet software package (which I guess JuWavelet is), but because it goes further to apply it to gravity wave measurements (as is mentioned even in the title) the manuscript should at least give some kind of overview of atmospheric gravity waves in modelling and observations and what parameters we are interested in. It's obvious to me as a gravity waves, or who might want to apply JuWavelet in other fields, and I think would strengthen the narrative of the paper. The authors could also briefly mention some other potential applications for JuWavelet beyond gravity waves, if they wish (I'm sure there are many possible uses in the geosciences).

- 3. Amplitude underestimation in the S-transform: As the authors know, for an infinitely long time series containing an infinitely long sinusoidal wave, the S-transform is able to measure the instantaneous amplitude of this wave perfectly. However, in the atmosphere gravity waves almost always occur in small packets containing maybe only a few wavecycles. In this case, when analysed with the ST, the convolution of the gravity wave packet's "envelope" with the Gaussian window in the ST results in an underestimation of the wave amplitude in the ST coefficients at that frequency due to spectral leakage, even if the wave is monochromatic. This underestimation is derived analytically for a hypothetical monochromatic gravity wave "packet" analysed by the 1-D, 2-D and 3-D S-transform in the Appendix A1 of Hindley et al. (2019). They found that for 1-D the effect is negligible, but it can be quite significant for higher dimensions. It can also be mitigated by adjusting the scaling parameter, similar to the authors' "free parameter k" (see below). Although the authors briefly mention this effect in Sect. 4.3, it would be really interesting for the authors to discuss (a) to what extent this amplitude underestimation happens in their application of the S-transform and (b) if they have any ideas or methods that they could use to counteract or avoid this underestimation in their approach. It's no problem that it occurs, it should just be mentioned in the paper a bit more clearly and I would be very interested to hear the authors comment on the issue and if they have any ideas or solutions.
- 4. How does JuWavelet deal with the ambiguity of wave directions in 2-D/3-D data? Specifically, when a wave packet has kx, ky and is ambiguous with a wave with -kx, -ky, what does JuWavelet measure? Does it limit only to analysing for angles of 0 to π or similar?

**Specific Comments**

- I.15 and elsewhere: "Stockwell transform" Minor point, but the S-transform is technically just called the S-transform (Stockwell et al., 1996), the S does not stand for Stockwell, strictly. But I appreciate the community often refers to it as such. Some journals even require it written S-transform using italic, based on their journal style.
- 2. I.15: "flavour" Some authors have debated over the years whether the S-transform (ST) is actually a modified type of CWT (Gibson et al., 2006), or whether is is a localised version of the Fourier transform, particularly in its discrete form (Stockwell, 2007). I have no feeling either way but the authors should mention that it has some concepts (like absolute referenced phase) which are quite different from a CWT, and that the ST "wavelet" does not have zero-mean so it is not strictly admissible in the CWT, but this is minor semantics. From a practical application point of view in the geosciences, one of the major differences between the ST and the CWT is that the coefficients of the ST can be directly interpreted as wave amplitude, whereas the CWT coefficients more closely resemble wavelet power (although I acknowledge this is mentioned later in the manuscript regarding Fig. 2). These are all subtleties and semantics, so we do not require a full exploration of these points, but the authors should ensure that they are consistent and mention some of the key differences that would be useful for the readers who might apply JuWavelet to their data.
- 3. I.19: The authors mention the 2-D S-transform of Hindley et al. (2016) but neglect the 3-D S-transform of Hindley et al. (2019), the code for which is actually N-dimensional and applies the 1-D, 2-D, 3-D and 4-D S-transform. I understand the narrative justification for JuWavelet though and there is definitely need for a multi-dimensional CWT/ST package in Python, but the authors could be more complete with their references to the literature. The authors should also probably mention the S3D method described by Lehmann et al. (2012) and Ern et al. (2017), if nothing else perhaps to simply avoid

forgetting their colleagues at Jülich! Actually, mentioning S3D works well in the narrative because JuWavelet overcomes the "cubes" limitations of S3-D and (possibly?!) the discretisation limitations of the *S*-transform of Hindley et al. (2019), but more on this below.

- 4. **I.40:** "atoms" not clear what atoms are in this context? Do they authors mean axioms? Even so that would not be quite accurate. Maybe just use "functions".
- 5. Sect. 3: "free parameter k" If I understand correctly, this is a very nice description by the authors of how (in their formulation) the S-transform basis functions are similar to the Morlet wavelet but with the free parameter k set to  $2\pi$ . It would be useful to mention, for consistency with Stockwell's original formulation (and to help readers who are less familiar!), that this is simply the same as scaling the Gaussian window in the ST with a standard deviation equal to 1/f, where f is the analysing frequency. For example, I'm not sure I understand correctly whether k is the value that scales the Gaussian window with frequency, or whether it is some multiple of that frequency? k also does not appear in the Eqn on 1.90 so perhaps it would be clearer for the reader if this was written explicitly, please rephrase.
- 6. Sect. 3: Further to the above point, it would be useful if the authors also discussed the effect of adjusting the ST's free parameter k by some multiple of  $2\pi$  to achieve improved spatial (spectral) localisation at the expense of spectral (spatial) localisation. Other studies have experimented with this depending on their geophysical dataset to achieve the desired results, such as Fritts et al. (1998); Pinnegar and Mansinha (2003); Hindley et al. (2016, 2019). Typically these studies describe this adjustment as a scaling parameter c such that the Gaussian window scales as c/f, referring to Stockwell's original formulation as mentioned in the previous comment. Does JuWavelet have this capability (I think it does) and did the authors experiment with it? In any case, it would be useful to mention this capability and when it might be applicable, to help users who may want to apply JuWavelet to their data. For example, Hindley et al. (2019) found that setting c = 1/4 (or c = 4, depending on whether the ST is written in the spatial or spectral or spectral domain) achieved improved spatial localisation in 3-D analysis of AIRS satellite observations of gravity waves. Because Hindley et al. (2019) only considered the dominant wave at each spatial location x, y, z, it did not matter very much if the spectral peak was broadened because the peak was still in the same location in the spectral domain  $f_x, f_y, f_z$ . Do the authors find the same with JuWavelet? It would be useful if the authors commented on this, and if they varied this free parameter when they analysed their 1-D, 2-D and 3-D datasets, and if they found improvements over previous methods, which I expect they do.
- 7. Sect. 4: Figure 3 is discussed in the text before Figure 2, consider rearranging.
- 8. Sect. 4.3: It would be nice to include a nod to Hindley et al. (2016) for Fig. 5, given the very close resemblance to their 2-D test case. It would also be nice to include a line plot of wavelength in versus wavelength out, and amplitude in versus amplitude out in order to assess the capability of JuWavelet for different wave scales in a dataset such as this. For example, Hindley et al. (2016) and Hindley et al. (2019) both showed that there is not a perfect 1 to 1 measurement for all wave scales for the ST in a test like this, but I would interested to see if the CWT mode of JuWavelet recovers the wavelengths perfectly, or if it has discretisation limitations like the Hindley et al. (2019) *S*-transform. If not, that would be a major strength.
- 9. I.160-161: This method of collapsing the 4-D spectrum by selecting the dominant frequency at each amplitude follows the approach of Hindley et al. (2016) and Hindley et al. (2019) for 6-D (also I.214), so it would be nice to include a reference because I don't recall seeing this method as common practise in the geosciences before these papers. Feel free to disagree, it's not essential.

- 10. **I.165-166:** Somewhat related to the point 6 above, the authors mention here that adjusting the Morlet parameter (is this k?) "can deliver more accurate amplitudes at the expense of spectral resolution". I'm not sure I fully understand the Morlet parameter then. I had naively thought that the setting  $k = 2\pi$  was equivalent to running the analysis in the S-transform mode (see the Eqn. on line 90), where the standard deviation of the analysing Gaussian window is  $\sigma = 1/f$ . I also hadn't considered that there could also be a scaling parameter in CWT mode that could adjust the spatial-spectral resolution, as is done for the ST. It would be very useful for the reader if the authors could explain this better, especially if a user is going to be using JuWavelet, they need to be aware of exactly how these options work.
- 11. I.167-169: I'm not it's enough to say that "the wavelengths/directions of the synthetic packets are well produced", the authors should be more quantitative. The easiest way is including a simple table or extra panel in Fig. 5 that shows the amplitude, wavelengths, phases, directions before and after the JuWavelet analysis.
- 12. **I.167-169:** Further on this point, as mentioned above does the JuWavelet formulation suffer from the discretisation limitations that one encounters when using e.g. the S-transform as derived from the discrete Fourier transform (DFT)? An S-transform based on the DFT (such as that first written in code by Stockwell) is very fast but is limited to discrete frequency voices, and might struggle to accurately measure low frequency waves like synthetic wave #8, which has a relatively large wavelength compared to the physical size of the image. Does JuWavelet also have this limitation, or is this solved by the derivation based on the CWT? If not, then this should be mentioned in the manuscript as it is a major advantage of JuWavelet (depending on the associated runtime).
- 13. Sect 4.4: Figure 7 is mentioned before Figure 6, please consider rearranging.
- 14. I.177: I'm intrigued by the *aspect* parameter. Is it not possible to achieve this result by setting the range of horizontal and vertical scales of the analysing wavelet (or scales and angles) to a given range, or does JuWavelet not have this functionality and it's better to stretch the data to get a more 1:1 aspect ratio of the waves? Also, does the stretching process actually resample the input data to make the JuWavelet input data larger and more "square", and does this have an effect on the runtime or memory requirements? As I said, I'm intrigued.
- 15. **I.180:** I'm not sure what is meant by "please see panel titles in App. B" and how this is related to the authors point. Do the authors expect the reader to run the code in order to read these I am guessing the panel titles they are referring to are  $f\lambda_x = \text{period[idx[0]]/np.cos(theta[idx[1]]):3.0fkm}$  and  $f\lambda_z = \text{period[idx[0]]/(cwt['aspect']*np.sin(theta[idx[1]])):3.1}fkm.$

I would say that having these written in Python in the Appendix is not sufficient for the reader in terms of explaining how to recover the horizontal and vertical wavelengths when setting  $aspect \neq 1$ . The authors should provide a clear equation in the text of how to recover these wavelengths when using different values of *aspect*, unless the JuWavelet package automatically calculates this? The reader should not have to either read or run the Python code to understand how this is done, it feels like the authors are cutting corners a little bit in the description. Maybe my Python is rusty but I also couldn't work out what the 3.0 and 3.1 were referring to.

- 16. **I.183:** "filtered" can the range of scales or angles in the CWT be filtered *before* the transform is computed to save runtime?
- 17. **I.187:** The authors should provide a little more information on this clustering algorithm, I think it's a very powerful addition to the software package to be able to detect e.g. the most important N waves

in a given image using this clustering algorithm, and could have widespread use in the geosciences. Just a little more information about which algorithm is used, how it is applied and what exactly it clusters scales/angles, amplitudes or both?

- 18. Sect. 4.5: Figure 7 is mentioned before Figure 6, consider rearranging.
- 19. Sect. 4.5: The use of a segmentation algorithm with JuWavelet to decompose, segment and then reconstruct the overlapping mountain waves is powerful and impressive! It would be interesting to apply this to the synthetic wave field in Fig. 5, or for cases where waves have more similar wavelengths than the mountain wave example shown, which seems a little easy. No need to include such an example, but a quick discussion of its strengths and limitations would be appreciated. On that point, what would happen if the the input data contained two gravity wave packets with very similar wavelengths and angles that were separated in distance and in opposite corners of the image? How would the segmentation algorithm cope with that case in the spectral domain? Would the two waves be recorded as one wave?
- 20. I.209-211: The azimuth and zenith angles seem quite coarse, is this to save runtime? Also, regarding the aspect ratio if one can specify the input scales and angles, why is the aspect ratio required to stretch the data in a given direction? I apologise if I have not understood correctly. Perhaps this could be clearer in the manuscript.
- 21. I.223: "can be simply realized" somehow I feel like parallelising the analysis of individual scales on the same input data like this is possible, but it might not be quite as "simple" as the authors suggest! It would interesting to include some kind of information about runtime for JuWavelet, although I appreciate this is relative to hardware and input. Even so, some ballpark numbers for runtime, or typical numbers of scales and angles, or some general advice on best practise would be appreciate for readers deciding if JuWavelet could work for their data. For example, should users always be prepared to have enough memory to generate a 6-D CWT/ST object if their input data is 3-D? This is important to know when analysing, for example, high resolution modelling output in 3-D. It would be useful there is an option to avoid the creating the 6-D object in memory and select only the dominant wave at each spatial location to output 3-D objects, for example.
- 22. **I.227:** "using the Morlet wavelet." I'm interested to know if the use of different wavelets, such as higher order complex Gaussian wavelets, could improve the amplitude estimation for wave packets in the JuWavelet CWT/ST analysis? No need to include this information in the manuscript, unless the authors are interested or they have some useful thoughts on the matter that might be worth including.
- 23. Appendix A: This may be more of a comment for the typesetting stage, but EGU journals are still two-column format as far I know. Therefore, the authors may struggle to show their 2-D and 3-D equations clearly, such as those on I.245-253. One solution could be to generalise the formulae to multiple dimensions by writing spatial and spectral vectors such as x, y, z and ωx, ωy, ωz as x = (x1, x2, ... xN) and ω = (ω1, ω2, ..., ωn), as was done in Hindley et al. (2019, their Eqns. 1 and 2) for an EGU journal. No problem if the authors are not concerned about the formatting, running the equation over multiple lines is also a solution.
- 24. **I.230:** This paper introduces the software package JuWavelet, but they don't actually mention where and how interested readers can download and use the package, or how it might be made available and under what conditions. In a method paper such as this, this information should definitely be included.

**Minor Typos and Suggestions**

- 1. I. 25 "brevity's sake" -> "brevity"
- 2. **I.33:** "precise" -> "(or more precisely, a function)"

**References**

- M. Ern, L. Hoffmann, and P. Preusse. Directional gravity wave momentum fluxes in the stratosphere derived from high-resolution airs temperature data. Geophy. Res. Lett., 44(1):475–485, 2017. doi: 10.1002/2016GL072007.
- D. C. Fritts, D. M. Riggin, B. B. Balsley, and R. G. Stockwell. Recent results with an mf radar at mcmurdo, antarctica: Characteristics and variability of motions near 12-hour period in the mesosphere. Geophy. Res. Lett., 25(3):297–300, 1998. ISSN 1944-8007. doi: 10.1029/97GL03702.
- P. C. Gibson, M. P. Lamoureux, and G. F. Margrave. Letter to the editor: Stockwell and wavelet transforms. Journal of Fourier Analysis and Applications, 12(6):713–721, 2006. ISSN 1069-5869. doi: {10.1007/s00041-006-6087-9}.
- N. P. Hindley, N. D. Smith, C. J. Wright, D. A. S. Rees, and N. J. Mitchell. A two-dimensional stockwell transform for gravity wave analysis of AIRS measurements. Atmospheric Measurement Techniques, 9(6): 2545–2565, June 2016. doi: 10.5194/amt-9-2545-2016.
- N. P. Hindley, C. J. Wright, N. D. Smith, L. Hoffmann, L. A. Holt, M. J. Alexander, T. Moffat-Griffin, and N. J. Mitchell. Gravity waves in the winter stratosphere over the southern ocean: high-resolution satellite observations and 3-d spectral analysis. Atmospheric Chemistry and Physics, 19(24):15377–15414, 2019. doi: 10.5194/acp-19-15377-2019.
- C. I. Lehmann, Y.-H. Kim, P. Preusse, H.-Y. Chun, M. Ern, and S.-Y. Kim. Consistency between fourier transform and small-volume few-wave decomposition for spectral and spatial variability of gravity waves above a typhoon. Atmos. Meas. Tech., 5(7):1637–1651, 2012. doi: 10.5194/amt-5-1637-2012.
- C. R. Pinnegar and L. Mansinha. The S-transform with windows of arbitrary and varying shape. Geophysics, 68(1):381–385, JAN-FEB 2003. doi: {10.1190/1.1543223}.
- R. G. Stockwell. A basis for efficient representation of the s-transform. Digital Signal Processing, 17(1):371 393, 2007. ISSN 1051-2004. doi: http://dx.doi.org/10.1016/j.dsp.2006.04.006.
- R. G. Stockwell, L. Mansinha, and R. P. Lowe. Localization of the complex spectrum: the s transform. IEEE Transactions on Signal Processing, 44(4):998–1001, Apr 1996. ISSN 1941-0476. doi: 10.1109/78.492555.

---

## Author Comment (AC1)

We thank the reviewers for their review, interesting questions, and detailed technical corrections. We took their suggestions and provided a significant revision of the submitted paper including the additional suggested information.

**1   Reply to Referee #1**

**1.1   Comments**

1. *First of all, I would like to congratulate the authors for developing a new and relevant tool for data analysis. Since this software package is freely available, many scientists will use it in the future. Consequently, the paper describing the application of this method needs to be more extensive and detailed. In particular, the paper needs to provide details on all the parameters employed (for example, s0, dj, js, jt, aspect, etc.). For instance, in chapter 4.4, the authors introduce an optional aspect parameter which is set to 40 in their example. How am I supposed to determine this coefficient if my data, for example, consists of time on the x-axis and altitude on the y-axis?*

   We thank the reviewer for this comments.

   We have carefully revised the application examples and provide more information on the set of used parameters as well as estimates on the required memory space and computation time (as reviewer #2 was asking for that). We also added a paragraph with a comprehensive explanation of the aspect parameter, which does not need to be used. W.r.t. the aspect parameter: the amplitude estimates become better if the analysed wavepackets are larger than the analysing basis functions. We found it useful to scale the axis of the data such that the analysed wave packets are squarish (which might differ for other types of wave, which extend longer than the few repetitions of atmospheric gravity waves).

2. *Essential to this work are not only well-presented examples, but also clear instructions on how to apply this method to any new data.*

   We added a new section describing in detail the 2-D decompose function call (the 1-D and 3-D functions can be used in a very similar way), its parameters and how to use it to analyse 2-D fields of arbitrary quantities and axis. We also expanded on the more general necessities of preparing a dataset for the analysis to prevent ringing and other common issues with frequency analysis. This should enable the application of the toolkit to other datasets.

3. *A plain text explaining the algorithm is also desirable.*

   We added a high-level description of the transforms, which are really just simple basis transformations. Also, the implementation section has now a paragraph providing a high-level description of the implementation.

   In particular, we greatly expanded the description of the effect on the configuration parameters and the similarities and differences of the Morlet-CWT and ST.

4. *Last but not least, an explanation of the results obtained is important. For example, Figure 7e shows some results of decomposition. The obtained horizontal wavelength is 381 km and vertical wavelength is 5.5 km. However, these parameters are not constant in time and altitude. Consider reconstructed fluctuations at 35 km altitude (Figure 1, blue line). Horizontal wavelength is shorter at the beginning than at the end. For orientation, a dashed line with $\lambda_x$=381 km is shown. In the range of 0 to approximately 500 km $\lambda_x$ ¡ 381 km and $\lambda_x$ ¿ 381 km after 500 km. At an altitude of 50 km ( compare orange and red dashed lines) $\lambda_x$ ¡ 381 km. Considering vertical cuts at 200 km and 600 km distances, an increasing $\lambda_z$ is noticeable above around 40 km and below around 30 km (see Figure 2). Figure 1. Perturbations at 35 km and 50 km altitudes are taken from Figure 7e of*

*the manuscript. Dashed lines indicate $\lambda_x$ =381 km. Figure 2. Perturbations at distances of 200 km and 600 km. Dashed lines indicate $\lambda_z$ =5.5 km. Waves with defined fixed parameters can be reconstructed using the inverse 2D-FFT. See examples in Fig. 3 c-f. A broader range of $\lambda_x$ and $\lambda_z$ is needed Figure 3. a,b: Figure7 (a, e) is taken from the manuscript. c-f: 2D-FFT reconstructed fluctuations with parameters as labeled in the titles. ) for the explanation of results from Figure 7 of the manuscript. Hopefully, after a clear description of the method, the interpretation of results will be more obvious. This comment addressed a specific example. The manuscript, however, requires the authors to offer a broader explanation of potential outcomes, demonstrating its applicability not just through examples but also for users working with different data.*

To avoid confusion between the subfigure heading and the depicted structure, we added points to indicate the center of the identified wave packet. The filtering employed is automatic and crude as seen be the spatial leakage. Also, the underlying wave structure is not synthetic and thus likely exhibits spatially varying frequencies, which further complicates the automatic separation. We enhanced the section with a discussion relating this effect.

Generally, we added more explanation of the involved major parameters such as scale $s$ and Morlet parameter $k$ to enable smart configuration choices by readers.

Specifically, we added a full new section discussing how to use the decomposition function with its associated parameters to get people started applying it to new data.

5. *Minor comment: Almost all examples shown in the manuscript can be found in the software package "juwavelet-v01.00.00/examples/", with the exception of Figures 6 and 8. Could these examples also be added to the repository?*

We added the missing examples to the repository. All figures can be produced now with the provided scripts.

**2   Reply to Referee #2**

**2.1   General Comments**

1. *Journal Scope: I notice that this article is submitted to Geophysical Model Development, which at first was a little surprising to me as it is perhaps better suited to Atmospheric Measurement Techniques? However, I will leave the editor to decide whether this is the right EGU journal and I certainly do not want to trigger another review cycle with AMT for the authors, as the paper is, in my view almost ready for acceptance.*

   We struggled with identifying the proper journal. However, as this paper describes a software and a modelling technique (i.e. approximating wave-like structures using wavelets), we decided for GMD as slightly better fitting.

   Also, while we use atmospheric gravity waves as example, the technique was partly developed by a seismologist and was also used in oceanography. We added more cites showcasing the use in different fields and switched the title to address waves in a generic fashion.

2. *The paper describes an analysis method specifically focused on the analysis of gravity waves in geophysical datasets, but doesn't actually mention what gravity waves are, how they are typically detected, why we would want to measure them, or any of the atmospheric science aspects of gravity wave remote sensing, detection or analysis. This is fine if the paper is only describing a generalised wavelet software package (which I guess JuWavelet is), but because it goes further to apply it to gravity wave measurements (as is mentioned even in the title) the manuscript should at least give some kind of overview of atmospheric gravity waves in modelling and observations and what parameters we are interested in. It's obvious to me as a gravity wave scientist but this information is essential for non-specialists who might not be familiar with gravity waves, or who might want to apply JuWavelet in other fields, and I think would strengthen the narrative of the paper. The authors could also briefly mention some other potential applications for JuWavelet beyond gravity waves, if they wish (I'm sure there are many possible uses in the geosciences).*

   We included a short paragraph on atmospheric gravity waves as suggested by the reviewer and point out why they must be analyzed using time-frequency analysis tools such as the CWT or the ST.

3. *Amplitude underestimation in the S-transform: As the authors know, for an infinitely long time series containing an infinitely long sinusoidal wave, the S-transform is able to measure the instantaneous amplitude of this wave perfectly. However, in the atmosphere gravity waves almost always occur in small packets containing maybe only a few wavecycles. In this case, when analysed with the ST, the convolution of the gravity wave packet's "envelope" with the Gaussian window in the ST results in an underestimation of the wave amplitude in the ST coefficients at that frequency due to spectral leakage, even if the wave is monochromatic. This underestimation is derived analytically for a hypothetical monochromatic gravity wave "packet" analysed by the 1-D, 2-D and 3-D S-transform in the Appendix A1 of Hindley et al. (2019). They found that for 1-D the effect is negligible, but it can be quite significant for higher dimensions. It can also be mitigated by adjusting the scaling parameter, similar to the authors' "free parameter k" (see below). Although the authors briefly mention this effect in Sect. 4.3, it would be really interesting for the authors to discuss (a) to what extent this amplitude underestimation happens in their application of the S-transform and (b) if they have any ideas or methods that they could use to counteract or avoid this underestimation in their approach. It's no problem that it occurs, it should just be mentioned in the paper a bit more*

*clearly and I would be very interested to hear the authors comment on the issue and if they have any ideas or solutions.*

We applied both the 2-D CWT and the 2-D ST to the synthetic wave field in Section 5.3 and both methods yield the same amplitudes. We also applied the 2-D CWT to the synthetic wave field using three different settings of the $k$ parameter (see Fig. 1). It becomes clear that amplitudes increase as $k$ becomes smaller due to the strong localization of the wave packets. Also, we notice more variability in wavelength and orientation as $k$ becomes smaller. A reconstruction by simply taking the real part of the complex-valued image containing the dominant coefficients is possible (Hindley et al., 2016). However, we note that with $k < 4$, a reconstruction as defined in Section 2 does not yield the original amplitude of the signal! This is important when dealing with clustering algorithms in the multi-dimensional spectral space. To guarantee invertability, we need $k > 4$. The caveat of this is an amplitude underestimation. We have not yet come up with an "amplitude-fix". One option might be an amplitude reallocation algorithm such as Synchrosqueezing. However, this is not in the scope of this work and we have not looked into that in detail.

4. *How does JuWavelet deal with the ambiguity of wave directions in 2-D/3-D data? Specifically, when a wave packet has $k_x$ , $k_y$ and is ambiguous with a wave with $-k_x$ , $-k_y$, what does JuWavelet measure? Does it limit only to analysing for angles of 0 to $\pi$ or similar?*

We thank the reviewer for this clarifying question. In 2-D, angles are in $[0, \pi]$, while in 3-D azimuth angles are in $[0, \pi]$ and zenith angles are in $[-\pi/2, \pi/2]$.

**2.2 Specific Comments**

1. *l.15 and elsewhere: "Stockwell transform" - Minor point, but the S-transform is technically just called the S-transform (Stockwell et al., 1996), the S does not stand for Stockwell, strictly. But I appreciate the community often refers to it as such. Some journals even require it written S-transform using italic, based on their journal style.*

We adopted the original style by Stockwell of $S$-transform and only noted the occasional naming of Stockwell transform in the main text.

2. *l.15: "flavour" - Some authors have debated over the years whether the S-transform (ST) is actually a modified type of CWT (Gibson et al., 2006), or whether is is is a localised version of the Fourier transform, particularly in its discrete form (Stockwell, 2007). I have no feeling either way but the authors should mention that it has some concepts (like absolute referenced phase) which are quite different from a CWT, and that the ST "wavelet" does not have zero-mean so it is not strictly admissible in the CWT, but this is minor semantics. From a practical application point of view in the geosciences, one of the major differences between the ST and the CWT is that the coefficients of the ST can be directly interpreted as wave amplitude, whereas the CWT coefficients more closely resemble wavelet power (although I acknowledge this is mentioned later in the manuscript regarding Fig. 2). These are all subtleties and semantics, so we do not require a full exploration of these points, but the authors should ensure that they are consistent and mention some of the key differences that would be useful for the readers who might apply JuWavelet to their data.*

We agree with the reviewers estimation.

All three transforms (Morlet-CWT, ST and Gabor-transform (i.e. a Short-time-Fourier-transform (STFT) with a Gaussian window) are mathematically very similar to the extent that the code implementing them in JuWavelet effectively supports all three and is identical with some small variations; albeit the Gabor-transform code path is not well tested

and inefficient in comparison to other widely available toolkits as in particular in electrical engineering, the STFT is a widely used tool. We neglected the Gabor transform to keep the paper concise, but will add a reference to make it more comprehensive to the reader.

3. *l.19: The authors mention the 2-D S-transform of Hindley et al. (2016) but neglect the 3-D S-transform of Hindley et al. (2019), the code for which is actually N-dimensional and applies the 1-D, 2-D, 3-D and 4-D S-transform. I understand the narrative justification for JuWavelet though and there is definitely need for a multi-dimensional CWT/ST package in Python, but the authors could be more complete with their references to the literature. The authors should also probably mention the S3D method described by Lehmann et al. (2012) and Ern et al. (2017), if nothing else perhaps to simply avoid forgetting their colleagues at Jülich! Actually, mentioning S3D works well in the narrative because JuWavelet overcomes the "cubes" limitations of S3-D and (possibly?!) the discretisation limitations of the S-transform of Hindley et al. (2019), but more on this below.*

The S3D transform is mathematically more akin to a Gabor transformation and thus only ancillary to this paper. We do not want to compare different analysis methods (in this case STFT and CWT), but provide a validated implementation of the CWT to the community. Certainly the Hilbert transform would need to be taken into account as well when going in this direction. As such, we decided to briefly mention recent advances in more specialized algorithms for gravity wave analysis, such as the S3D as well as the Hilbert-transform based Unified Wave Diagnostics.

We are again sorry for being imprecise with our language. We cited the use of 3-D CWT by Hindley et al. without specifically highlighting its application for 3-D analysis, though. Later in the text we were largely referring to the ever decreasing readily availability of *code* with increasing dimensionality, a gap we wanted to close here. For example, 2-D code has been described in the given reference (and is also available in expensive commercial toolkits), whereas we are mostly aware of very few 3-D applications beyond the work of Hindley et al. (2019).

We rephrased the section to make our meaning w.r.t. to readily available and free implementation more clear and added an explicit mentioning of the 3-D CWT use by Hindley et al. (2019).

4. *l.40: "atoms" - not clear what atoms are in this context? Do they authors mean axioms? Even so that would not be quite accurate. Maybe just use "functions".*

We replaced the term with "basis functions".

5. *Sect. 3: "free parameter k" - If I understand correctly, this is a very nice description by the authors of how (in their formulation) the S-transform basis functions are similar to the Morlet wavelet but with the free parameter k set to $2\pi$. It would be useful to mention, for consistency with Stockwell's original formulation (and to help readers who are less familiar!), that this is simply the same as scaling the Gaussian window in the ST with a standard deviation equal to $1/f$, where $f$ is the analysing frequency. For example, I'm not sure I understand correctly whether $k$ is the value that scales the Gaussian window with frequency, or whether it is some multiple of that frequency? $k$ also does not appear in the Eqn on l.90 so perhaps it would be clearer for the reader if this was written explicitly, please rephrase.*

We expanded the section "Morlet wavelet and S-transform" further.

We now provide also the integrals in real space, which were previously neglected as they bear no importance for the implementation (previous focus of the paper), but which explain the transforms much better. Here, we can and do discuss the relations between the Gaussian envelope, the scale parameters, and the harmonic term. This allows to further highlight the similarities and differences of the transformations.

Indeed for $k = 2\pi$, the standard deviation of the Gaussian envelope agrees with the scale parameter and the analysing period length.

6. *Sect. 3: Further to the above point, it would be useful if the authors also discussed the effect of adjusting the ST's free parameter k by some multiple of $2\pi$ to achieve improved spatial (spectral) localisation at the expense of spectral (spatial) localisation. Other studies have experimented with this depending on their geophysical dataset to achieve the desired results, such as Fritts et al. (1998); Pinnegar and Mansinha (2003); Hindley et al. (2016, 2019). Typically these studies describe this adjustment as a scaling parameter c such that the Gaussian window scales as c/f, referring to Stockwell's original formulation as mentioned in the previous comment. Does JuWavelet have this capability (I think it does) and did the authors experiment with it? In any case, it would be useful to mention this capability and when it might be applicable, to help users who may want to apply JuWavelet to their data. For example, Hindley et al. (2019) found that setting c = 1/4 (or c = 4, depending on whether the ST is written in the spatial or spectral or spectral domain) achieved improved spatial localisation in 3-D analysis of AIRS satellite observations of gravity waves. Because Hindley et al. (2019) only considered the dominant wave at each spatial location x, y, z, it did not matter very much if the spectral peak was broadened because the peak was still in the same location in the spectral domain $f_x$, $f_y$, $f_z$. Do the authors find the same with JuWavelet? It would be useful if the authors commented on this, and if they varied this free parameter when they analysed their 1-D, 2-D and 3-D datasets, and if they found improvements over previous methods, which I expect they do.*

JuWavelet does have the capability to freely choose the $k$ parameter, which is noted in the paper, but not prominently enough. We will expand this and add a discussion on the motivation why one would want to do so, and also give some additional cites to literature, which are certainly useful for the intended audience.

Doing new research is out-of-scope of this paper, though; the whole research area of varying $k$, maybe even use a non-Gaussian window, and the related effects on estimating amplitudes (and maybe even the definition of amplitudes of wave packets) is very interesting but beyond the scope of this paper. We hope that a readily available and simply modifiable Python implementation allows more people to perform studies and advance the field.

7. *Sect. 4: Figure 3 is discussed in the text before Figure 2, consider rearranging.*

We changed the order of the figures as suggested.

8. *Sect. 4.3: It would be nice to include a nod to Hindley et al. (2016) for Fig. 5, given the very close resemblance to their 2-D test case. It would also be nice to include a line plot of wavelength in versus wavelength out, and amplitude in versus amplitude out in order to assess the capability of JuWavelet for different wave scales in a dataset such as this. For example, Hindley et al. (2016) and Hindley et al. (2019) both showed that there is not a perfect 1 to 1 measurement for all wave scales for the ST in a test like this, but I would interested to see if the CWT mode of JuWavelet recovers the wavelengths perfectly, or if it has discretisation limitations like the Hindley et al. (2019) S-transform. If not, that would be a major strength.*

We sincerely apologize for that oversight and included a "nod" to Figure 2 of Hindley et al. (2016). We altered Figure 5 to even closer resemblance and added not only the reconstruction of the dominant modes but also as suggested by the reviewer a line plot to see how well input and out wavelengths and orientations agree. As described in the text, we find a non-negligible difference in input and output wavelength for the two larger scale wave packets which we assume is due to their strong spatial localization with respect to their wavelength and due to their proximity to the domain's boundary.

9. *l.160-161: This method of collapsing the 4-D spectrum by selecting the dominant frequency at each amplitude follows the approach of Hindley et al. (2016) and Hindley et al. (2019) for 6-D (also l.214), so it would be nice to include a reference because I don't recall seeing this method as common practise in the geosciences before these papers. Feel free to disagree, it's not essential.*

Again, we apologize for giving not enough credit at this point to the work of Hindley et al. We included a reference accordingly.

10. *l.165-166: Somewhat related to the point 6 above, the authors mention here that adjusting the Morlet parameter (is this k?) "can deliver more accurate amplitudes at the expense of spectral resolution". I'm not sure I fully understand the Morlet parameter then. I had naively thought that the setting $k = 2\pi$ was equivalent to running the analysis in the S-transform mode (see the Eqn. on line 90), where the standard deviation of the analysing Gaussian window is $\sigma = 1/f$. I also hadn't considered that there could also be a scaling parameter in CWT mode that could adjust the spatial-spectral resolution, as is done for the ST. It would be very useful for the reader if the authors could explain this better, especially if a user is going to be using JuWavelet, they need to be aware of exactly how these options work.*

This strikes a similar point as a comment above. We have expanded the mathematical description by the real-space integrals, we allow better to see the similarities between the Morlet-CWT and the ST and how the $k$ parameter affects both, including the relationship to the standard deviation.

In effect, the $k$ parameter varies the modulation frequency, not the width of the Gaussian, but the result is almost identical. The added complication of this approach is a difference between the "scaling" parameter of the CWT and the "period" of the analysed frequency, which is compensated for by JuWavelet as it delivers the proper wavelength for each scale parameter.

We add the wavelet analysis of the synthetic wave field using $k = 2, 4, 2\pi$ in Figure 1. It becomes clear that amplitudes increase as $k$ decreases while wavelengths and orientations become more variable which is to be expected due to the coarser spectral resolution for smaller $k$ values. However, with $k = 2$ the admissibility condition is violated and a reconstruction would yield higher amplitudes than the original wave field.

11. *l.167-169: I'm not it's enough to say that "the wavelengths/directions of the synthetic packets are well produced", the authors should be more quantitative. The easiest way is including a simple table or extra panel in Fig. 5 that shows the amplitude, wavelengths, phases, directions before and after the JuWavelet analysis.*

As mentioned above, we included a line plot to show how well input and output wavelengths and orientations agree (at least at the central position of the respective wave packet).

12. *l.167-169: Further on this point, as mentioned above does the JuWavelet formulation suffer from the discretisation limitations that one encounters when using e.g. the S-transform as derived from the discrete Fourier transform (DFT)? An S-transform based on the DFT (such as that first written in code by Stockwell) is very fast but is limited to discrete frequency voices, and might struggle to accurately measure low frequency waves like synthetic wave #8, which has a relatively large wavelength compared to the physical size of the image. Does JuWavelet also have this limitation, or is this solved by the derivation based on the CWT? If not, then this should be mentioned in the manuscript as it is a major advantage of JuWavelet (depending on the associated runtime).*

We have added a sentence mentioning that the ST as originally described by Stockwell employs only a finite number of dyadic scales identical to the length of the analysed data vector, similar to a more common FFT. However, his ST could have been computed for finer

grids as well, similar to the implementation of JuWavelet, which draws heavily on Torrence's 1-D implementation. In either case, the majority of the employed runtime is caused by the inverse FFT necessary for each scale, i.e. the runtime of the algorithm increases linearly with the number of analysed scales.

We are aware of the efficient use of filter banks for discrete orthogonal and biorthogonal wavelet analysis, which allow for faster analysis and reconstruction. While CWT and DWT are intimately connected, they are still conceptually quite different beasts. The implementation here focuses on the CWT and is thus not able to employ some of the optimizations of the discrete wavelet transform.

To make this explicit, we added a section to the implementation section analysing the algorithmic complexity of the transforms.

13. *Sect 4.4: Figure 7 is mentioned before Figure 6, please consider rearranging.*

We changed the order of the figures as suggested.

14. *l.177: I'm intrigued by the aspect parameter. Is it not possible to achieve this result by setting the range of horizontal and vertical scales of the analysing wavelet (or scales and angles) to a given range, or does JuWavelet not have this functionality and it's better to stretch the data to get a more 1:1 aspect ratio of the waves? Also, does the stretching process actually resample the input data to make the JuWavelet input data larger and more "square", and does this have an effect on the runtime or memory requirements? As I said, I'm intrigued.*

Since wavelet angles are equally distributed in $[0, \pi]$ it is best to scale the wave field to be analyzed in such a way that wavefront directions are distributed isotropically. We achieve this by adjusting the aspect parameter. A short paragraph is added to the manuscript. This could simply be achieved by supplying "wrong" sampling distances to the transform, but the computed periods would be wrong. The aspect parameter allows for the use of proper units (if possible). No resampling is necessary or performed. In other terms, one can interpret this as the wavelet itself being compressed/stretched in one spatial dimension.

15. *l.180: I'm not sure what is meant by "please see panel titles in App. B" and how this is related to the authors point. Do the authors expect the reader to run the code in order to read these - I am guessing the panel titles they are referring to are [. . . (skipped code)] I would say that having these written in Python in the Appendix is not sufficient for the reader in terms of explaining how to recover the horizontal and vertical wavelengths when setting aspect ! = 1. The authors should provide a clear equation in the text of how to recover these wavelengths when using different values of aspect, unless the JuWavelet package automatically calculates this? The reader should not have to either read or run the Python code to understand how this is done, it feels like the authors are cutting corners a little bit in the description. Maybe my Python is rusty but I also couldn't work out what the 3.0 and 3.1 were referring to.*

As the reviewer point this out, we realize that this is indeed silly.

Aside changing the software to directly provide the directional wavelengths (which are indeed not trivial to compute, even without the aspect ratio) we added a paragraph detailing the feature in the method part:

An additional feature only available for the 2-D and 3-D transformations is the addition of an *aspect* ratio, which effectively scales the last dimension before applying the transformation. The basis functions of the CWT and ST are defined to be isotropic. If the analysis shall be employed using actual units, e.g. kilometres, this can pose a problem for vertical cross-sections of atmospheric gravity waves, which are often hundreds times wider than tall and often only have one or two

periods. Using a simple 2-D ST will use basis functions, which like extend vertically beyond the wave structure and thus cause an amplitude dampening. Scaling the field vertically such that the measured waves are more "quadratic" or "cubic", respectively resolves this issue. JuWavelet is capable of scaling the last dimension by a scalar; this has no effect on the mathematical transform itself, however, the computation of directional wavelengths changes. For this reason, the corresponding directional wavelengths in the two or three dimensions are provided for the higher order transformations.

16. *l.183: "filtered" - can the range of scales or angles in the CWT be filtered before the transform is computed to save runtime?*

The routines providing only the dominant coefficients had such filtering options. We extended this now to have this option consistently for all decomposition routines. The normal decompositions use regular arrays for providing the result, which we fill with a NaN value for skipped coefficients.

17. *l.187: The authors should provide a little more information on this clustering algorithm, I think it's a very powerful addition to the software package to be able to detect e.g. the most important N waves in a given image using this clustering algorithm, and could have widespread use in the geosciences. Just a little more information about which algorithm is used, how it is applied and what exactly it clusters - scales/angles, amplitudes or both?*

We agree that a powerful clustering algorithm would be of great benefit to the community, but the one provided within the JuWavelet package is not that algorithm. It is contained as it demonstrates the capabilities of the transform and to provide a starting point for further work. For scientific work, we used the output of the algorithm as a starting point but filtered manually on top. We hope that the software package gains traction and that contributions such as a well-examined clustering algorithm will be added in the future. Still, we expanded the description of the algorithm:

Due to the finite spectral and spatial resolution of the employed basis, the basis function closest in parameters to the true wave packet will typically have the largest coefficient, but spectrally neighbouring basis functions will still have large values decreasing with "distance" in the wavelet space. Including these in the reconstruction is important to retain the amplitude of the packet. The algorithm assumes that overlapping wave packets are separable in spectral space by coefficients below a configurable threshold. Due to the high dimensionality of the CWT coefficients, this is a reasonable assumption. The algorithm then identifies the largest coefficient not part of an identified cluster and first identifies the scales and angles associated with this cluster by looking for neighbouring scales and angles at this point above the threshold. In a second step, the spatial extent is explored in the same way, starting from the identified scales and angles. The algorithm repeats until no further cluster can be constructed from remaining coefficients.

18. *Sect. 4.5: Figure 7 is mentioned before Figure 6, consider rearranging.*

We changed the order of the figures as suggested.

19. *Sect. 4.5: The use of a segmentation algorithm with JuWavelet to decompose, segment and then reconstruct the overlapping mountain waves is powerful and impressive! It would be interesting to apply this to the synthetic wave field in Fig. 5, or for cases where waves have more similar wavelengths than the mountain wave example shown, which seems a little easy. No need to include such an example, but a quick discussion of its strengths and limitations would be appreciated. On that point, what would happen if the the input data contained two gravity wave packets*

*with very similar wavelengths and angles that were separated in distance and in opposite corners of the image? How would the segmentation algorithm cope with that case in the spectral domain? Would the two waves be recorded as one wave?*

As suggested by the reviewer we applied the watershed segmentation algorithm to the 4-D wavelet power spectrum of the 2-D synthetic wave field in Section 4.3 and reconstructed all eight wave packets individually. We include a Figure in the supplements. Only two wave packets that are spatially and spectrally too close could not be separated and are identified as one wave packet. That might also answer the reviewer's second question. As long as the watershed segmentation algorithm identifies a minimum in the wavelet power spectrum-no matter in which dimension-it will separate the 4-D CWT.

20. *l.209-211: The azimuth and zenith angles seem quite coarse, is this to save runtime? Also, regarding the aspect ratio - if one can specify the input scales and angles, why is the aspect ratio required to stretch the data in a given direction? I apologise if I have not understood correctly. Perhaps this could be clearer in the manuscript.*

The simple and honest answer is: yes, we choose only so few angles in order to save runtime and memory. As mentioned in the manuscript, we are planning to implement parallelization to shorten the computation runtime. The default setting of the code creates $jt$ azimuth and $jp$ zenith angles with constant increments. Changing the aspect ratio is a quick way to change the increments' scaling.

21. *l.223: "can be simply realized" - somehow I feel like parallelising the analysis of individual scales on the same input data like this is possible, but it might not be quite as "simple" as the authors suggest! It would interesting to include some kind of information about runtime for JuWavelet, although I appreciate this is relative to hardware and input. Even so, some ballpark numbers for runtime, or typical numbers of scales and angles, or some general advice on best practise would be appreciate for readers deciding if JuWavelet could work for their data. For example, should users always be prepared to have enough memory to generate a 6-D CWT/ST object if their input data is 3-D? This is important to know when analysing, for example, high resolution modelling output in 3-D. It would be useful there is an option to avoid the creating the 6-D object in memory and select only the dominant wave at each spatial location to output 3-D objects, for example.*

We thank the reviewer for this comment. Indeed, in 3-D memory space and runtime blow up. We now state in the manuscript that the 3-D wave field has $250 \times 300 \times 40$ entries and takes up $11.4\,\mathrm{MB}$. The 3-D CWT takes $0.74\,\mathrm{h}$ of computation time using 16 physical cores (32 logical CPUs via 2-way hyperthreading) and $64\,\mathrm{GB}$ of RAM with 16 scales, 6 azimuth angles and 7 zenith angles. The 6-D array of complex-valued wavelet coefficients takes up $13.5\,\mathrm{GB}$ of disc space.

22. *l.227: "using the Morlet wavelet." - I'm interested to know if the use of different wavelets, such as higher order complex Gaussian wavelets, could improve the amplitude estimation for wave packets in the JuWavelet CWT/ST analysis? No need to include this information in the manuscript, unless the authors are interested or they have some useful thoughts on the matter that might be worth including.*

We do not have a wavelet available that would have better properties. It is likely already optimal in many characteristics due to its construction. In particular atmospheric gravity waves are a very difficult beast to analyse as they have often so few repetitions. But the question is very interesting indeed, so we give some of our thoughts on that matter:

We believe that an amplitude underestimation will be an inherent property of any CWT. The ST is highly tuned to reproduce correct amplitudes for waves of infinite extent. Naturally, if a wave packet of finite extent is analysed, the provided coefficients will decrease, notably at

that if the extent of the packet is getting as small as the "meaningful" extent of the Gaussian window. It is likely that specific wavelets are subject to less dampening for small wave packets and we are aware of some research to that matter. The whole affair is problematic if only due to the fact that the amplitude of a very small wave packet is not well-defined in particular taking the uncertainty principle into account.

We think that it might be more worthwhile to characterize the dampening effect for certain wave packet sizes and to correct for it in a post-processing step taking the spatial extent of the identified waves into account; there are some papers on this topic, e.g. combining the CWT with a Hilbert transform, which we find to be out-of-scope for this paper. Secondary, an even more generous analysis operator than the CWT might be able to derive more information from the field. The CWT and ST might be hindered in their power by their invertibility (which we believe is a key feature of this technique; if this is not required, the S3D might indeed be the more pragmatic approach even though it lacks a similar strong mathematical background as the CWT).

However, either endeavour (which have been tried with varying success before) is beyond the scope of this paper, which mostly tries to proof the mathematical correctness of our Python-based open-source implementation of the Morlet-CWT and ST to the scientific community.

23. *Appendix A: This may be more of a comment for the typesetting stage, but EGU journals are still two-column format as far I know. Therefore, the authors may struggle to show their 2-D and 3-D equations clearly, such as those on l.245-253. One solution could be to generalise the formulae to multiple dimensions by writing spatial and spectral vectors such as x, y, z and $\omega_x$ , $\omega_y$ , $\omega_z$ as $x = (x_1$ , $x_2$ , $\dots x_N$ ) and $\omega = (\omega_1$ , $\omega_2$ , $\cdots$ , $\omega_n$ ), as was done in Hindley et al. (2019, their Eqns. 1 and 2) for an EGU journal. No problem if the authors are not concerned about the formatting, running the equation over multiple lines is also a solution.*

We are thankful for the helpful suggestions and will tackle these issues in collaboration with the copy-editor, once the paper is accepted.

24. *24. l.230: This paper introduces the software package JuWavelet, but they don't actually mention where and how interested readers can download and use the package, or how it might be made available and under what conditions. In a method paper such as this, this information should definitely be included.*

We added URLs and an example installation command using the pip tool in addition to the Zenodo citation (which is, effectively a DOI providing a downloadable and citable version of the software).

[Figure]

Figure 1: Results of the 2-D CWT applied to the synthetic wave field from Section 5.3 with $k = 2, 4, 2\pi$. First row shows dominant amplitudes, second row shows associated wavelengths, and third row shows associated orientations.

**References**

Hindley, N. P., Smith, N. D., Wright, C. J., Rees, D. A. S., and Mitchell, N. J.: A two-dimensional Stockwell transform for gravity wave analysis of AIRS measurements, Atmos. Meas. Tech., 9, 2545–2565, https://doi.org/10.5194/amt-9-2545-2016, 2016.